# Recurrent Space-time Graph Neural Networks

**Andrei Nicolicioiu**[*] **Iulia Duta**[*]
Bitdefender, Romania
`anicolicioiu, iduta@bitdefender.com`

**Marius Leordeanu**
Bitdefender, Romania
Institute of Mathematics of the Romanian Academy
University "Politehnica" of Bucharest
`marius.leordeanu@imar.ro`

## Abstract

Learning in the space-time domain remains a very challenging problem in machine learning and computer vision. Current computational models for understanding spatio-temporal visual data are heavily rooted in the classical single-image based paradigm. It is not yet well understood how to integrate information in space and time into a single, general model. We propose a neural graph model, recurrent in space and time, suitable for capturing both the local appearance and the complex higher-level interactions of different entities and objects within the changing world scene. Nodes and edges in our graph have dedicated neural networks for processing information. Nodes operate over features extracted from local parts in space and time and over previous memory states. Edges process messages between connected nodes at different locations and spatial scales or between past and present time. Messages are passed iteratively in order to transmit information globally and establish long range interactions. Our model is general and could learn to recognize a variety of high level spatio-temporal concepts and be applied to different learning tasks. We demonstrate, through extensive experiments and ablation studies, that our model outperforms strong baselines and top published methods on recognizing complex activities in video. Moreover, we obtain state-of-the-art performance on the challenging Something-Something human-object interaction dataset.

## 1 Introduction

Video data is available almost everywhere. While image level recognition is better understood, visual learning in space and time is far from being solved. The main challenge is how to model interactions between objects and higher level concepts, within the large spatio-temporal context. For such a difficult learning task it is important to efficiently model the local appearance, the spatial relationships and the complex interactions and changes that take place over time.

Often, for different learning tasks, different models are preferred, such that they capture the specific domain priors and biases of the problem [1]. Convolutional neural networks (CNNs) are preferred on tasks involving strong local and stationary assumptions about the data. Recurrent models are chosen when data is sequential in nature. Fully connected models could be preferred when there is no known structure in the data. Our recurrent neural graph efficiently processes information in both space and time and can be applied to different learning tasks in video.

We propose Recurrent Space-time Graph (RSTG) neural networks, in which each node receives features extracted from a specific region in space-time using a backbone deep neural network.

---

[*]Equal contribution.

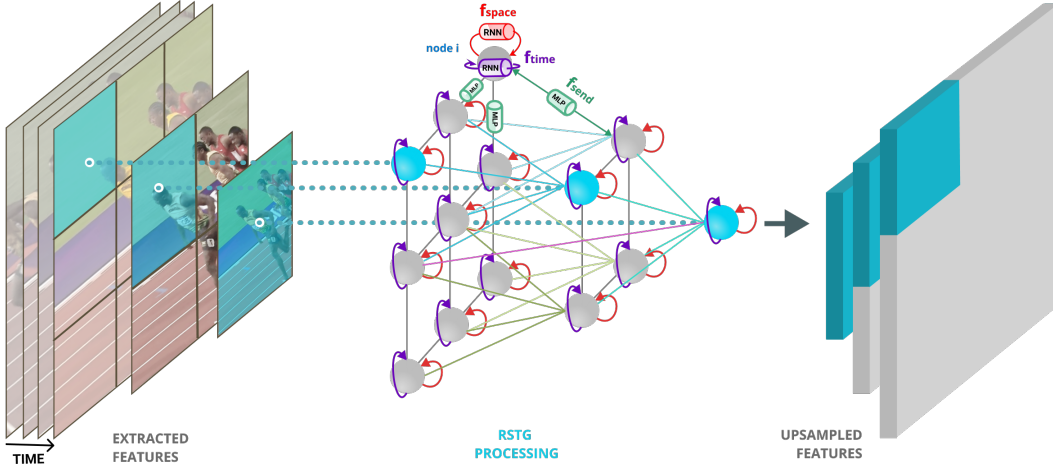

Figure 1: The RSTG-to-map architecture: the input to RSTG is a feature volume, extracted by a backbone network, down-sampled according to each scale. Each node receives input from a cell, corresponding to a region of interest in space. The edges between different nodes represent messages in space, the red links are spatial updates, while the purple links represent messages in time. All the extracted (input to graph) and up-sampled features (output from graph) have the same spatial and temporal dimension $T \times H \times W \times C$ and are only represented at different scales for a better visualisation.

Global processing is achieved through iterative message passing in space and time. Spatio-temporal processing is factorized, into a space processing stage and a time processing stage, which are alternated within each iteration. We aim to decouple, conceptually, the data from the computational machine that processes the data. Thus, our nodes are processing units that receive inputs from several sources: local regions in space at the present time, their neighbor spatial nodes as well as their past memory states (Fig. 1).

**Main contributions.** We sum up our contributions into the following three main ideas:

1. We propose a **novel computational model** for learning in spatio-temporal domain. Space and time are treated differently, while they function together in complementary ways. Our model is **general** and could be applied to various learning problems. It could also be used as **a processing block** in combination with other powerful models.

2. We **factorize space and time** and **process them differently** within a unified neural graph model from an unstructured video. In extensive ablation studies we show the importance of each graph component and also demonstrate that different temporal and spatial processing is crucial for learning in space-time domain. Through **recurrent** and factorized space-time processing our model achieves a relatively **low computational complexity**.

3. We introduce a new synthetic dataset, with complex interactions, to analyse and evaluate different spatio-temporal models. We obtain a performance that is superior to several powerful baselines and top published methods. More importantly, we obtain **state-of-the-art** results on the challenging Something-Something, real world dataset.

**Relation to previous work:** Iterative graph based methods have a long history in machine learning and are currently enjoying a fast-growing interest [1, 2]. Their main paradigm is the following: at each iteration, messages are passed between nodes, information is updated at each node and the process continues until convergence or a stopping criterion is met. Such ideas trace back to work on image denoising, restoration and labeling [3, 4, 5, 6], with many inference methods, graphical models and mathematical formulations being proposed over time for various tasks [7, 8, 9, 10, 11, 12, 13].

Current approaches combine the idea of message passing between graph nodes, from graphical models, with convolution operations. Thus, the idea of graph convolutions was born. Initial methods generalizing conv nets to the case of graph structured data [14, 15, 16] learn in the spectral domain of the graph. They are approximated [17] by message passing based on linear operations [18] or

MLPs [19]. Aggregation of messages needs permutation invariant operators such as max or sum, the last one being proved superior in [20], with attention mechanism [21] as an alternative.

Recurrence in graph models has been proposed for sequential tasks [22, 23] or for iteratively processing the input [24, 25]. Recurrence is used in graph neural nets [22] to tackle symbolic tasks with single input and sequential language output. Different from them, we have two types of recurrent stages, with distinct functionality, one over space and the other over time.

The idea of modeling complex, higher order and long range spatial relationships by the spatial recurrence relates to more classical work using pictorial structures [26] to model object parts and their relationships and perform inference through iterative optimization algorithms. The idea of combining information at different scales also relates to classic approaches in object recognition, such as the well-known spatial pyramid model [27, 28].

Long-range dependencies in sequential language are captured in [29] with a self-attention model. It has a stack of attention layers, each with different parameters. It is improved in [24] by performing operations recurrently. This is similar to our recurrent spatial processing stage. As mentioned before, our model is different by adding another complementary dimension - the temporal one. In [25] new information is incorporated into the existing memory by self-attention using a temporary new node. Then each node is updated by an LSTM [30]. Their method is applied on program evaluation, simulated environments used in reinforcement learning and language modeling where they do not have a spatial dimension. Their nodes act as a set of memories. Different from them, we receive new information for each node and process them in multiple interleaved iterations of our two stages.

Initial node information could come from each local spatio-temporal point in convolutional feature maps [31, 32] or from features corresponding to entities detected by external methods, such as objects [33, 34, 35] or skeletons [36]. Also, the approach in [37] is to extract objects and form relations between objects from pairs of time steps randomly chosen. Different from that methods, our nodes are not attached to specific volumes in time and space. Also, we do not need pre-trained higher-level detectors, our model working on unstructured videos.

While the above methods need access to the whole video at test time, ours is recurrent and can function in an online, continuous manner in time. All space-time positions in the input volume are connected in [32, 33, 38]. In contrast, we treat space and time differently and prove the effectiveness of our choice in experiments. A 1D convolution is used in [36] to temporally connect only the nodes corresponding to the same skeleton joint and recently [34] send messages between nodes corresponding to the same entities, while we recurrently update in time the state of each node. We could see our different handling of time and space as an efficient factorization into simpler mechanisms that function together along different dimensions. The work in [39, 40] confirm our hypothesis that features could be more efficiently processed by factorization into simpler operations. The models in [41, 42, 43] factorize 3D convolutions into 2D spatial and 1D temporal convolutions.

For spatio-temporal processing, some methods, which do not use explicit graph modeling, encode frames individually using 2D convolutions and aggregate them in different ways [44, 45, 46]; others form relations as functions (MLPs) over sets of frames [47] or use 3D convolution inflated from existing 2D convolutional networks [48] . Optical flow could be used as input to a separate branch of a 2D ConvNet [49] or used as part of the model to guide the kernel of 3D convolutions [50]. To cover both spatial and temporal dimensions simultaneously, Convolutional LSTM [51] can be used, augmented with additional memory [52] or self-attention in order to update LSTM hidden states [53].

## 2 Recurrent Space-time Graph Model

The Recurrent Space-time Graph (RSTG) model is designed to process data in both space and time, to capture both local and long range spatio-temporal interactions (Fig. 1). RSTG takes into consideration local information by computing over features extracted from specific locations and scales at each moment in time. Then it integrates long range spatial and temporal information by iterative message passing at the spatial level between connected nodes and by recurrence in time, respectively. The space and time message passing is coupled with the two stages succeeding one after another.

Our model takes a video and process it using a backbone function into a features volume $F \in \mathbb{R}^{T \times H \times W \times C}$, where $T$ is the time dimension and $H$,$W$ the spatial ones. The backbone function could be modeled by any deep neural network that operates over single frames or over

**Algorithm 1** Space-time processing in RSTG model.

**Input:** Time-space features $F \in \mathbb{R}^{T \times H \times W \times C}$

**repeat**
    $\mathbf{v}_i \leftarrow extract\_features(F_t, i)$         $\forall i$

    **for** $k = 0$ **to** $K - 1$ **do**

        $\mathbf{v}_i = \mathbf{h}_i^{t,k} = f_{time}(\mathbf{v}_i, \mathbf{h}_i^{t-1,k})$   $\forall i$
        $\mathbf{m}_{j,i} = f_{send}(\mathbf{v}_j, \mathbf{v}_i)$       $\forall i, \forall j \in \mathcal{N}(i)$
        $\mathbf{g}_i = f_{gather}(\mathbf{v}_i, \{\mathbf{m}_{j,i}\}_{j \in \mathcal{N}(i)})$   $\forall i$
        $\mathbf{v}_i = f_{space}(\mathbf{v}_i, \mathbf{g}_i)$         $\forall i$

    **end for**

    $\mathbf{h}_i^{t,K} = f_{time}(\mathbf{v}_i, \mathbf{h}_i^{t-1,K})$         $\forall i$
    $t = t + 1$

**until** end-of-video

$\mathbf{v}_{final} = f_{aggregate}(\{\mathbf{h}_i^{1:T,K}\}_{\forall i})$

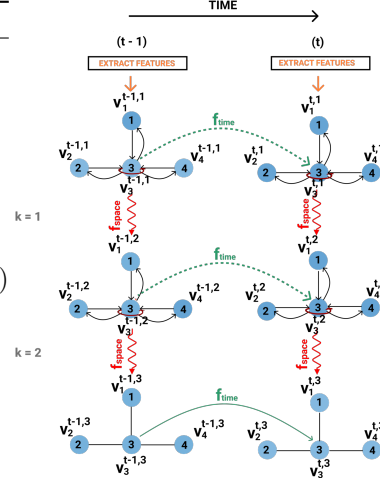

Figure 2: Two Space Processing Stages ($K = 2$) from top to bottom, each one preceded by a Temporal Processing Stage.

space-time volumes. Thus, we extract local spatio-temporal information from the video volume and we process it using our graph, sequentially, time step after time step. This approach makes it possible for our graph to also process a continuous flow of spatio-temporal data and function in an online manner.

Instead of fully connecting all positions in time and space, which is costly, we establish long range interactions through recurrent and complementary Space and Time Processing Stages. Thus, in the temporal processing stage, each node receives a message from the previous time step. Then, at the spatial stage, the graph nodes, which now have information from both present and past, start exchanging information through message passing. Space and time are coupled and performed alternatively: after each space iteration $iter$, another time iteration follows, with a message coming from past memory associated with the same space iteration $iter$. The processing stages of our algorithm are succinctly presented in Alg. 1 and Fig. 2. They are detailed below. The code for the full model can be found in our repository [2].

**Graph Creation.** We create N nodes connected in a graph structure and use them to process a features volume $F \in \mathbb{R}^{T \times H \times W \times C}$. Each node receives input from a specific region (a window defined by a location and scale) of the features volume at each time step $t$ (Fig. 1). At each scale we downsample the $H \times W$ feature maps into $h \times w$ grids, each cell corresponding to one node. Two nodes are connected if they are neighbours in space or if their regions at different scales intersect.

## 2.1 Space Processing Stage

Spatial interactions are established by exchanging messages between nodes. The process involves 3 steps: **send** messages between all connected nodes, **gather** information at node level from the received messages and **update** internal nodes representations. Each step has its own dedicated MLP. Message passing is iterated $K$ times, with time processing steps followed by space processing steps, at each iteration.

**Message sending function.** A given message between two nodes should represent relevant information about their pairwise interaction. Thus, the message is a function of both the source and destination nodes $j$ and $i$, respectively. The function, $f_{send}(\mathbf{v}_j, \mathbf{v}_i)$ is modeled as a multilayer perceptron (MLP) applied on the concatenation of the two node features:

[2] https://github.com/IuliaDuta/RSTG

$$f_{send}(\mathbf{v}_j, \mathbf{v}_i) = \text{MLP}_s([\mathbf{v}_j | \mathbf{v}_i]) \in \mathbb{R}^D. \tag{1}$$

$$\text{MLP}_a(\mathbf{x}) = \sigma(W_{a_2}\sigma(W_{a_1}(\mathbf{x}) + b_{a_1}) + b_{a_2}). \tag{2}$$

**Position-aware messages.** The pairwise interactions between nodes should have positional awareness - each node should be aware of the position of the neighbor that sends a particular message. Therefore we include the position information as a (linearized) low-resolution $6 \times 6$ map in the message body sent with $f_{send}$, by concatenating the map to the rest of the message. The actual map is formed by putting ones for the cells corresponding to the region of interest of the sending nodes and zeros for the remaining cells, and then applying filtering with a Gaussian kernel.

**Gather function.** Each node receives a message from each of its neighbours and aggregates them using the $f_{gather}$ function, which could be a simple sum of all messages or an attention mechanism that gives a different weight to each message, according to its importance. In this way, a node could choose what information to receive. In our implementation, the attentional weight function $\alpha$ is computed as the dot product between features of the two nodes, measuring their similarity.

$$f_{gather}(\mathbf{v}_i) = \sum_{j \in \mathcal{N}(i)} \alpha(\mathbf{v}_j, \mathbf{v}_i) f_{send}(\mathbf{v}_j, \mathbf{v}_i) \in \mathbb{R}^D. \tag{3}$$

$$\alpha(\mathbf{v}_j, \mathbf{v}_i) = (W_{\alpha_1}\mathbf{v}_j)^T (W_{\alpha_2}\mathbf{v}_i) \in \mathbb{R}. \tag{4}$$

**Update function.** We update the representation of each node with the information gathered from its neighbours, using function $f_{space}$ modeled as a multilayer perceptron (MLP). We want each node to be capable of taking into consideration global information while also maintaining its local identity. The MLP is able to combine efficiently new information received from neighbours with the local information from the node's input features.

$$f_{space}(\mathbf{v}_i) = \text{MLP}_u([\mathbf{v}_i | f_{gather}(\mathbf{v}_i)]) \in \mathbb{R}^D. \tag{5}$$

In general, the parameters $W_u, b_u$ could be shared among all nodes at all scales or each set could be specific to the actual scale.

## 2.2 Time Processing Stage

Each node updates its state in time by aggregating the current spatial representation $f_{space}(\mathbf{v}_i)$ with its time representation from the previous step using a recurrent function. In order to model more expressive spatio-temporal interactions and to give it the ability to reason about all the information in the scene, with knowledge about past states, we put a Time Processing Stage before each Space Processing Stage, at each iteration, and another Time Processing Stage after the last spatial processing.

Thus messages are passed iteratively in both space and time, alternatively. The Time Processing Stage at iteration $k$ updates each node's internal state $v_i^{t,k}$ with information from its corespondent state $v_i^{t-1,k}$, at iteration $k$, in the previous time $t-1$, resulting in features that take into account both spatial interactions and history (Fig. 2).

$$\mathbf{h}_{i,time}^{t,k} = f_{time}(\mathbf{v}_{i,space}^k, \mathbf{h}_{i,time}^{t-1,k}). \tag{6}$$

## 2.3 Aggregation step

The aggregation $f_{aggregate}$ function could produce two types of final representations, a 1D vector or a 3D map. In the first case, denoted **RSTG-to-vec**, we obtain the vector encoding by summing the representation of all the nodes from the last time step. In the second case, denoted **RSTG-to-map**, we create the inverse operation of the node creation, by sending the processed information contained in each node back to the original region in the space-time volume as shown in Figure 1. For each scale, we have $h * w$ nodes with $C$-channel features, that we arrange in a $h \times w$ grid resulting in a volume of size $h \times w \times C$. We up-sample the grid map for each scale into $H \times W \times C$ maps and sum all maps for all scales for the final $H \times W \times C$ representation.

Table 1: Accuracy on SyncMNIST dataset, showing the capabilities of different parts of our model.

| Model | 3 SyncMNIST | 5 SyncMNIST |
|---|---|---|
| Mean + LSTM | 77.0 | - |
| Conv + LSTM | 95.0 | 39.7 |
| I3D | - | 90.6 |
| Non-Local | - | 93.5 |
| RSTG: Space-Only | 61.3 | - |
| RSTG: Time-Only | 89.7 | - |
| RSTG: Homogenous | 95.7 | 58.3 |
| RSTG: 1-temp-stage | 97.0 | 74.1 |
| RSTG: All-temp-stages | **98.9** | 94.5 |
| RSTG: Positional All-temp | - | **97.2** |



Figure 3: On each row we present frames from videos of 5SyncM-NIST dataset. In each video sequence two digits follow the exact same pattern of movement. The correct classes: "3-9" "6-7" and "9-1".

## 2.4 Computational complexity

We analyse the computational complexity of the RSTG model. If $N$ is the number of nodes in a frame and $E$ the number of edges, we have $O(2E)$ messages per space-processing stage, as there are two different spatial messages in each edge direction. With a total of $T$ time steps and $K$ (=3) spatio-temporal message passing iterations, each of the $K$ spatial message passing iterations is preceded by a temporal iteration, resulting in a total complexity of $O(T \times (2E) \times K + T \times N \times (K+1))$. Note that $E$ is upper-bounded by $N(N-1)/2$. Without the factorisation, with messages between all the nodes in time and space (similar to [32, 33]), we would arrive at a complexity of $O(T^2 \times N^2 \times K)$ in the number of messages, which is quadratic in time. Note that our lower complexity is due to the recurrent nature of our model and the space-time factorization.

## 3 Experiments

We perform experiments on two video classification tasks, which involve complex object interactions. We experiment on a video dataset that we create synthetically, containing complex patterns of movements and shapes, and on the challenging Something-Something-v1 dataset, involving interactions between a human and other objects [54].

### 3.1 Learning patterns of movements and shapes

There are not many available video datasets that require modeling of difficult object interactions. Improvements are often made by averaging the final predictions over space and time [38]. The complex interactions and the structure of the space-time world still seem to escape the modeling capabilities. For this reason, and to better understand the role played by each component of our model in relation to some very strong baselines, we introduce a novel dataset, named SyncMNIST.

We make several MNIST digits move in complex ways. We designed the dataset such that the relationships involved are challenging in both space and time. The dataset contains $600K$ videos showing multiple digits, where all of them move randomly, apart from a pair of digits that moves synchronously - that specific pair determines the class of the activity pattern, for a total of $45$ unique digit pairs (classes) plus one extra class (no pair is synchronous).

In order to recognize the pattern, a given model has to reason about the location in space of each digit, track them across the entire time in order to learn the association between a label and a pair of digits that moves synchronously. The data has $18 \times 18$ size digits moving on a black $64 \times 64$ background for 10 frames. In Fig. 3 we present frames from three different videos used in our experiments. We trained and evaluated our models first on an easier 3 digits (3SyncMNIST) dataset and then, only the best models were trained and tested on the harder 5 digits dataset (5SyncMNIST).

We compared against four strong baseline models that are often used on video understanding tasks. For all tested models we used a convolutional network as a backbone. It is a small CNN with 3 layers,

pre-trained to classify a digit randomly placed in a frame of the video. It is important to notice that published models such as MeanPooling+LSTM, Conv+LSTM, I3D and Non-Local, have the same ranking on our SyncMNIST dataset as on other datasets such as UCF-101 [55], HMDB-51 [56], Kinetics (see [48]) and Something-Something (see [33]). The available performance of these models on all datasets can be found in Section A of the Appendix.

It is also important that the performance of different models seems to be well correlated with the ability of a specific model to incorporate and process time axis. This aspect, combined with the fact that, by design, on SyncMNIST the temporal dimension is important, make the tests on SyncMNIST relevant.

**Mean pooling + LSTM:** Use backbone for feature extraction, spatial mean pool and temporally aggregate them using an LSTM. This model is capable of processing information from distant time-steps but it has poor understanding of spatial information.

**ConvNet + LSTM:** Replace the mean pooling with convolutional layers that are able to capture fine spatial relationships between different parts of the scene. Thus, it is fully capable of analysing the entire video, both in space and in time.

**I3D:** We adapt the I3D model [48] with a smaller ResNet [57] backbone to maintain the number of parameters comparable to our model. 3D convolutions are capable of capturing some of the longer range relationships both spatially and temporally.

**Non-Local:** We used the previous I3D architecture as a backbone for a Non-Local [32] model. We obtained best results with one non-local block in the second residual block.

**Implementation details for RSTG:**   Our recurrent neural graph model (RSTG) uses the initial 3-layer CNN as backbone, an LSTM with $512$ hidden state size for the $f_{time}$ and RSTG-to-vec as aggregation. We use 3 scales with $1 \times 1$, $2 \times 2$ and $3 \times 3$ grids with nodes of dimension $512$. We implement our model in Tensorflow framework [58]. We use cross-entropy as loss function and trained the model end-to-end with SGD with Nesterov Momentum with value $0.9$ for momentum, starting from a learning rate of $0.0001$ and decreasing by a factor of 10 when performance saturates.

In Table 3 results show that RSTG is significantly more powerful than the competitors. Note that the graph model runs on single-image based features, without any temporal processing at the backbone level. The only temporal information is transmitted between nodes at the higher graph level.

### 3.1.1   Ablation study

Solving the moving digits task requires a model capable of capturing pairwise interactions both in space and time. RSTG is able to accomplish that, through spatial connections between nodes and the temporal updates of their state. In order to prove the benefits of each element, we perform experiments that shows the contributions brought by each one and present them in Table 3. We observed the efficiently transfer capabilities of our model between the two versions of the SyncMNIST dataset. When pretrained on 3SyncMNIST, our best model RSTG-all-temp-stages achieves $90\%$ of its maximum performance in a number of steps in which an uninitialized model only attains $17\%$ of its maximum performance.

**Space-Only RSTG:** We create this model in order to prove the necessity of having powerful time modeling. It performs the Space Processing Stage on each frame, but ignores the temporal sequence, replacing the recurrence with an average pool across time dimension, applied for each node. As expected, this model obtains the worst results because the task is based on the movement of each digit, an information that could not be inferred only from spatial exploration.

**Time-Only RSTG:** This model performs just the Time Processing Stage, without any message-passing between nodes. The features used in the recurrent step are the initial features extracted from the backbone neural network, which takes as input single frames.

**Homogeneous Space-time RSTG:** This model allows the graph to interact both spatially and temporally, but learn the same set of parameters for the MLPs that compute messages in time and space. Thus, time and space are computed in the same way.

Table 2: Comparison with state-of-the-art models on Something-Something-v1 dataset showing Top-1 and Top-5 accuracy.

| Model | Backbone | Val Top-1 | Val Top-5 |
|---|---|---|---|
| C2D | 2D ResNet-50 | 31.7 | 64.7 |
| TRN [47] | 2D Inception | 34.4 | - |
| ours C2D + RSTG | 2D ResNet-50 | **42.8** | **73.6** |
| MFNet-C50 [59] | 3D ResNet-50 | 40.3 | 70.9 |
| I3D [33] | 3D ResNet-50 | 41.6 | 72.2 |
| NL I3D [33] | 3D ResNet-50 | 44.4 | 76.0 |
| NL I3D + Joint GCN [33] | 3D ResNet-50 | 46.1 | 76.8 |
| ECO$_{Lite-16F}$ [60] | 2D Inc+3D Res-18 | 42.2 | - |
| MFNet-C101 [59] | 3D ResNet-101 | 43.9 | 73.1 |
| I3D [42] | 3D Inception | 45.8 | 76.5 |
| S3D-G [42] | 3D Inception | 48.2 | 78.7 |
| ours I3D + RSTG | 3D ResNet-50 | **49.2** | **78.8** |

**Heterogeneous Space-time RSTG:** We developed different schedulers for our spatial and temporal stages. In the first scheduler, used in the **1-temp RSTG** model, for each time step, we performed 3 successive spatial iteration, followed by a single final temporal update. The second scheduler, the **all-temp RSTG** model, alternates between the spatial and temporal stages (as presented in Alg.1). We use one Time Processing Stage before each of the three Space-Processing Stages, and a last Time Processing Stage to obtain the final nodes representation.

**Positional All-temp RSTG:** This is the previous all-temp RSTG model, but enriched with positional embeddings used in $f_{send}$ function as explained in Section 2. This model, which is our best and final model, is also able to reason about global locations of the entities.

## 3.2    Learning human-object interaction

In order to evaluate our method in a real world scenario involving complex interactions, we use the Something-Something-v1 dataset [54]. It consists of a collection of 108499 videos with 86017, 11522 and 10960 videos for train, validation and test splits respectively. It has 174 classes for fine-grained interactions between humans and objects. It is designed such that classes can be discriminated not by some global context or background but from the actual specific interactions.

For this task we investigate the performance of our graph model combined with two backbones, a 2D convolutional one (C2D [32]), based on ResNet-50 architecture and an I3D [48] model inflated also from the ResNet-50. We start with backbones pretrained on Kinetics-400 [48] dataset as provided by [32] and train the whole model end-to-end.

We analyse our both aggregation types, described in Section 2.3. For RSTG-to-vec we use the last convolutional features given by the I3D backbone as input to our graph model and obtain a vector representation. To facilitate the optimisation process we use residual connections in RSTG, by adding the results of the graph processing to the pooled features of the backbone. For the second case we use intermediate features of I3D as input to the graph and also add them to the graph output by a residual connection and continue the I3D model. For this purpose we need both the input and the output of the graph to have the same dimension. Thus we use RSTG-to-map to obtain a 3D map at each time step.

**Training and evaluation.**    For training, we uniformly sample 32 frames from each video resized such the height is 256, preserving the aspect ratio and randomly cropped to a $224 \times 224$ clip. For inference, we apply the backbone fully convolutional on a $256 \times 256$ crop with the graph taking features from larger activation maps. We use 11 square clips uniformly sampled on the width of the frames for covering the entire spatial size of the video, and use 2 samplings along the time dimension. We mean pool the clips output for the final prediction.

Figure 4: We show running time (clips / s) on the left axis and final accuracy on the right axis.

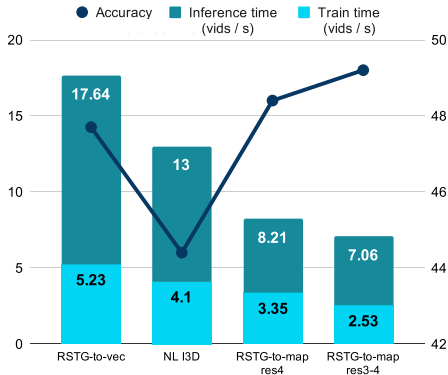

Table 3: Ablation study showing where to place the graph inside the ResNet-50 I3D backbone. For our best model we use two different graphs after the res3 and res4 stages of the I3D.

| Model | Top-1 | Top-5 |
|---|---|---|
| RSTG-to-vec | 47.7 | 77.9 |
| RSTG-to-map res2 | 46.9 | 76.8 |
| RSTG-to-map res3 | 47.7 | 77.8 |
| RSTG-to-map res4 | 48.4 | 78.1 |
| RSTG-to-map res3-4 | **49.2** | **78.8** |

**Results.** We analyse how our graph model could be used to improve I3D by applying RSTG-to-map at different layers in the backbone and RSTG-to-vec after the last convolutional layer. In all cases the model achieves competitive results, and the best performance is obtained using the graph in the res3 and res4 blocks of the I3D as shown in Table 3. We compare against recent methods on the Something-Something-v1 dataset and show the results in Table 2. Among the models using 2D ConvNet backbones, ours obtains the best results (with a significant improvement of more than $8\%$ over all methods using a 2D backbone, for the Top-1 setup). When using the I3D backbone, RSTG reaches state-of-the-art results, with $1\%$ improvement over all methods (Top-1 case) and $3.1\%$ improvement over top methods (Top-1 case) with the same 3D-ResNet-50 backbone.

**Computational requirements** We show the compute times for different variants of our model and for the Non-Local model using the Resnet-50 backbone on Something-Something videos running on one Nvidia GTX 1080 Ti GPU in Figure 4. We observe that our RSTG-to-vec model is faster, while having better accuracy than the Non-Local model, whereas our top performing model RSTG-to-map res3-4 further increase the results at the cost of being about 2x slower than RSTG-to-vec. Our RSTG-to-vec requires 6.95 GB for training and 1.23 GB for inference, while RSTG-to-map res3-res4 requires 7.50 GB and 1.93 GB respectively, with a batch of 2 clips.

## 4   Conclusions

In this paper we introduce the Recurrent Space-time Graph (RSTG) neural network model, which is specifically designed to learn efficiently in space and time. The graph, at each moment in time, starts by receiving local space-time information from features produced by a given backbone network. Then it moves towards global understanding by passing messages over space between different locations and scales and recurrently in time, by having a different past memory for each space-time iteration. Our model is unique in the literature in the way it processes space and time, with several main contributions: 1) it treats space and time differently; 2) it factorizes them and uses recurrent connections within a unified neural graph model from an unstructured video, with relatively low computational complexity; 3) it is flexible and general, being relatively easy to adapt to various learning tasks in the spatio-temporal domain; 4) our ablation study justifies the structure and different components of our model, which obtains state-of-the-art results on the challenging Something-Something dataset. In future work we plan to further study and extend our model to other higher-level tasks such as semantic segmentation in spatio-temporal data and vision-to-language translation.

**Acknowledgements:** This work has been supported in part by Bitdefender and UEFISCDI, through projects EEA-RO-2018-0496 and TE-2016-2182.

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
