[Supplementary Material]

# Appendix: Recurrent Space-time Graph Neural Networks

**Andrei Nicolicioiu,**[*] **Iulia Duta**[*]
Bitdefender, Romania
`anicolicioiu, iduta@bitdefender.com`

**Marius Leordeanu**
Bitdefender, Romania
Institute of Mathematics of the Romanian Academy
University "Politehnica" of Bucharest
`marius.leordeanu@imar.ro`

## A  Models ranking on 5SyncMNIST vs other video datasets

We present in Figure 1 the available results of our RSTG model and the published models MeanPool-ing+LSTM, Conv+LSTM, I3D and Non-Local on UCF-101 [1], HMDB-51 [2], Kinetics (see [3]), Something-Something (see [4]) and on our 5SyncMNIST dataset. There is one curve per dataset, with one point on the curve per method, shown in increasing order of performance, which is preserved across datasets. As seen by the strictly increasing lines, the same rank order of all the models is maintained on several datasets, including ours. This affirms the consistent behaviour of the methods as well as the relevance of the datasets.

Figure 1: Performance of different models on several datasets

## B  Details about using the RSTG module together with a backbone

**Feature extraction:**  In the following section we give some additional technical details of the way the RSTG model is designed in order to work in conjunction with a backbone. More specifically, we discuss how we combine RSTG with I3D as the backbone model. Our model is suited for being inserted at multiple layers of any backbone network. We test the model when it is included after a

---

[*]Equal contribution.

Table 1: Architecture of our RSTG-to-map-res4 model, with 4 scales, that processes features from the res4 stage of the I3D ResNet-50 model.

| model | layer | output size |
|---|---|---|
| | input | $32 \times 224 \times 224 \times 3$ |
| I3D | conv1 | $32 \times 112 \times 112 \times 64$ |
| | pool1 | $32 \times 56 \times 56 \times 64$ |
| | res2 | $32 \times 56 \times 56 \times 256$ |
| | pool2 | $16 \times 56 \times 56 \times 256$ |
| | res3 | $16 \times 28 \times 28 \times 512$ |
| | res4 | $16 \times 14 \times 14 \times 1024$ |
| RSTG | Graph creation | $16 \times 4 \times 4 \times 512$ <br> $16 \times 3 \times 3 \times 512$ <br> $16 \times 2 \times 2 \times 512$ <br> $16 \times 1 \times 1 \times 512$ |
| | $\begin{bmatrix} \text{Temporal Processing Stage} \\ \text{Spatial Processing Stage} \end{bmatrix} \times 3$ | |
| | Temporal Processing Stage | $16 \times 30 \times 512$ |
| | Up-sample each grid <br> $1 \times 1 \times 1$ conv | $16 \times 14 \times 14 \times 512$ <br> $16 \times 14 \times 14 \times 2048$ |
| I3D | res5 | $16 \times 14 \times 14 \times 2048$ |
| | mean pool, fc | $1 \times 1 \times 1 \times 174$ |

single I3D stage or after multiple such stages. This insertion is done as follows (as seen in Figure 1 in the main paper): we take the output from a specific backbone layer, having dimension $\mathbb{R}^{T \times H \times W \times C}$ and use it as an input to our graph model. For example, if we give as video input 32 frames of size $224 \times 224$ frames to I3D and use the features given by the res4 stage (as input to our RSTG model), we obtain a $16 \times 14 \times 14 \times 2048$ input feature map as shown in Table 1.

**Node creation:** As shown in Figure 1 in the main paper, we pool features at $S$ different scales, each corresponding to regions in the original image, ranging from areas covering parts of the image to areas covering the entire image. For each of the $S$ scales, we down-sample the features into increasingly smaller maps (one for each scale). Each of these maps form an $M \times M$ grid, each point representing a node. Then, at each time step, each node receives a temporal slice from the features corresponding to its cell in the grid.

**Output creation:** We process the graph with our interleaved Space and Time Processing Stages. The recurrent Time Processing Stage runs for $T$ steps, with each node internal state $h_i^t$ at each time step having an increasingly better temporal information during this process.

By using a residual connection, we add the graph features to the backbone features, thus they must have the same number of channels and temporal dimension. For this, we first project each node back to the initial $2048$ dimension and aggregate together the graph and backbone features.

In the case of RSTG-to-vec models, we sum all the nodes and add them to the global spatial mean pooling of the backbone and average them across time.

In the case of RSTG-to-map models, we also need to have the same spatial dimension for graph and backbone features. For each scale, the node features, arranged as grids, are up-sampled to the original input spatial size. This is done for each of the $S$ scales, resulting in $S$ $16 \times 14 \times 14 \times 2048$ maps that are then summed together into a single map of the same size $16 \times 14 \times 14 \times 2048$. The resulting map is further summed with the backbone input features map using a residual connection. The final map thus obtained (which is represented by the up-sampled output features in Figure 1 of the main paper) is then fed into the remaining stages of the I3D model to obtain the final prediction.

**Experiments on the graph position inside the backbone:** We conduct several experiments showing different ways of combining our RSTG model with the I3D backbone, by varying which I3D layers are used as input to the graph (res2, res3, res4), the final aggregation methods (RSTG-to-vec, RSTG-to-map) and the number of graphs used between different I3D layers.

We observe that our graph model obtains the best results with higher level features from res4 stage, while it obtains the wors results using lower level res2 stage features. The best results are achieved when two graphs are stacked with input features from different layers of the backbone (i.e. res3 and res4).

## C  Additional experiments regarding the form of adjacency matrix

In order to validate our choice of connectivity used in the spatial processing, we experimentally test two types of adjacency matrix. The first is formed as stated in the main paper by connecting two nodes if they are neighbours in space or if their regions at different scales intersect. The second is a matrix full of ones, forming a completely connected graph. We observe that the sparser version used in the rest of the paper obtains better results.

Table 2: Results of RSTG-to-vec model obtained by varying the adjacency matrix.

| Model | Top-1 | Top-5 |
|---|---|---|
| RSTG-to-vec - sparse | 47.7 | 77.9 |
| RSTG-to-vec - full | 46.9 | 76.8 |

## D  Details about the baselines used in SyncMNIST experiments

We offer a more detailed description of the baselines used in the experiments on SyncMNIST dataset.

**Mean pooling + LSTM:** We use as backbone a 3-layer CNN to independently extract features from each frame. Each convolutional layer uses $3 \times 3$ filters followed by ReLU non-linearity and $2 \times 2$ max-pooling. We aggregate the spatial information using a final global average pooling, and use an LSTM with $512$ hidden state dimension to process all frame features into a feature vector used to make the final prediction. This model is capable of processing information from distant time-steps but it has poor understanding of spatial information because of the lost information due to the pooling.

**ConvNet + LSTM:** We include an additional ConvNet on top of the previous CNN backbone. It consists of a three convolutional layers, with stride 1, without pooling, with ReLU non-linearity and batch normalization [5] between them. For the second baseline, the extra convolutional layers (without spatial pooling and no down-sizing of the activation maps) are able to capture fine spatial relationships between different parts of the scene. The features from the last layer are also passed through the same LSTM model. Thus, the second baseline is fully capable of analyzing the entire video, both in space and in time.

**I3D:** We used a version of I3D [3] with inflated 3D convolutions adapted from ResNet-50. We considered only 3 residual stages (res2, res3, res5) with fewer number of blocks (2 for each residual stage) and fewer filters such that the number of parameters became comparable with our models (around 10M). While the 3D convolutions process local space-time volumes at a time, they are very powerful and capable to capture some of the longer range relationships through repeated convolutions at multiple layers of depth.

**Non-Local:** We used the previous I3D architecture (described as I3D above) as a backbone for a Non-Local[6] model. We tried adding multiple blocks at different stages, and obtained the best results with one non-local block in the second residual stage.

## E  An intuitive view of Recurrent Space-time Graph Neural Networks

The experiments presented in the paper strongly indicate that the RSTG graph structure, which operates iteratively and recurrently over space and time brings an important additional value to the

Figure 2: Sampled frames from our 5SyncMNIST dataset. With green arrows we show the path of the pair of digits that move synchronously and determine the class label of the video (for a total of 45+1 possible class labels, where the is no pair of synchronously moving digits); with red arrows we show the paths of other digits that move randomly, independently from the pair that move synchronously.

convolutional network backbone, whether that backbone has 3D or 2D convolutions. We believe that the main reason for this fact is hidden in the way objects and events, which are more distant in space and time, interact and influence each other to determine a specific action or activity. Complex activities are often composed of many events, which in turn are defined by several interactions between objects that take place at different positions, scales and moments in time. It is also important that the arrangement of events in time is different, conceptually, than the arrangements of objects in space.

The ideas above suggest that there is a need for a computational structure that is able to process information locally but is also able to quickly send the results of such computations to distant regions in space and time. What the recurrent space-time graph model has over the more uniform and local convolutional backbone networks is, first and foremost, its ability to separate conceptually the local computation at the level of nodes from the passing of messages between nodes at the level of space-time edges. Then, the message passing routine, which is iterative, can quickly send information globally and reach a convergent state that puts in agreement the local computations.

In the case of convolutional networks, the spreading of information from the local to the global levels seems to be done less efficiently, through many layers of processing, in a more continuous and local manner, in which time and space are treated more or less in the same uniform fashion. The RSTG graph structure treats from the start time and space differently. It encourages the computation at the node level to reach an agreement by passing messages between nodes for several iterations, in space and also in time (from past state to the current one). We believe that this iterative process is suited for such higher levels of abstraction in order to learn efficiently about how objects interact to form first simpler and more local events and then more complex and more global activities.

These concluding remarks are supported by our experiments in which we show that RSTG brings a stronger boost over a powerful backbone model when operating over the higher level features provided by this model. The results suggest that the RSTG graph adds complementary capabilities to the input network, by being able to capture, perhaps more efficiently, the discrete-continuous and complex structure of the space-time world at higher levels of abstraction.