[Reviews · NeurIPS 2019]

Reviewer 1



The paper is well-written and easy to follow. technically it seems to be correct and more importantly, the method is novel. this paper had the highest quality in my batch and I enjoyed reading it. the problem is well-defined and well-motivated, the distinction from other related work is clear, the solution is intuitive and novel and it works on a real dataset. weaknesses are listed in section 5.

Reviewer 2



• This work claims to be the first to do space-time factorization in neural graph processing. However, [A][B] use similar space-time factorization, in which separate spatial and temporal graph convolutions are performed. Considering this, the novelty of this work is weakened. • The actor type, as well as the spatial layout in both datasets used, are relatively rigid. More experiments on complex human-object interaction datasets, e.g., Charades, would be helpful in showing the scalability of the adopted rigid region-split scheme. It would also be helpful to compare with the existing space-time graphical modeling approaches, e.g., [B][C][33], on such datasets. • Compared with previous space-time video modeling works, this work is different mainly in two components: one is to use message passing instead of graph convolution; and the other is the rigid region-split scheme compared with the explicit object detection manner. There are no ablation studies concerning these two modules to shed a light on which part actually brings the performance boost. • How is the number of scales determined? Studies for analyzing the correlation between performance and number of scales are also missing. [A] Spatial Temporal Graph Convolutional Networks for Skeleton-Based Action Recognition. Yan et al. AAAI 2018. [B] Stacked Spatio-Temporal Graph Convolutional Networks for Action Segmentation. Ghosh et al. Arxiv preprint. [C] Video Relationship Reasoning using Gated Spatio-Temporal Energy Graph. Tsai et al. CVPR 2019. ------------------------- After rebuttal: Thanks to the authors for the comprehensive response. However it could have been better if the authors provided in the rebuttal the results that they promised to include in the final paper. I stand to my previous decision of "above acceptance threshold."

Reviewer 3



1. Although the space-time graph is not a very novel, however, the proposed recurrent space-time graph is more principled than previous GCN applied on videos. It fits with a network rather being an indepent module. 2. The paper is easy to read and understand. 3. The paper has the potential to be applied widely in video processing community, conditioned on its computational efficiency. 4. I found no obvious faults in this paper. It is evaluated properly.

[Author Response · NeurIPS 2019]

We thank the reviewers for their comments and suggestions, which will help us better present our work.

**In response to Reviewers 1 and 3 regarding the computational cost and comparison to NL model:**

| Model | Train ↑ (vids/s) | Inference ↑(vids/s) | Accuracy ↑ |
|---|---|---|---|
| RSTG-to-vec | **5.23** | **17.64** | 47.7 |
| NL-I3D | 4.10 | 13.00 | 44.4 |
| RSTG-to-map res4 | 3.35 | 8.21 | 48.4 |
| RSTG-to-map res3-4 | 2.53 | 7.09 | **49.2** |

We show the runtimes for different variants of our model and the NL-I3D model using the Resnet-50 backbone on Something-Something videos. Times are similar: our RSTG-to-vec model is the fastest and has better accuracy than the NL model, while our top performing model RSTG-to-map res3-4 is about 2x slower than RSTG-to-vec. We will include the comparisons in the camera ready, if accepted.

**In response to Reviewers 1 and 2 regarding experiments on Charades dataset:**

We agree that Charades represents a good dataset for evaluation. In the paper, considering time and computational
resources constraints, we tested on two datasets. Something-Something is a large-scale, real-world dataset (newer
and 10x larger than Charades), in which complex interactions in space and time are more relevant than specific object
classes. Next we will perform experiments on Charades and present them in future work.

**Additional responses to Reviewer 1:**

*More detailed analysis and discussion:* We thank the reviewer for this suggestion. We will do our best to improve
such analysis and discussion in the camera ready, if accepted.

*Computational cost and results on Charades:* Please see above our answers to Reviewers regarding computational
cost as well as results on Charades. We will include computation times in the final version.

*Claim L212-214 clarification:* We thank the reviewer for this suggestion. We will either remove the claim or clarify it with the following experimental evidence, space permitting. What we mean is that the methods we compared against maintained the same rank order on several datasets, including ours. This affirms the consistent behaviour of the methods as well as the relevance of the datasets. In the Figure we plot the performances on different datasets, as reported in [33, 45]. There is one curve per dataset, with one point on the curve per method, shown in increasing order of performance, which is preserved across datasets.

**Additional responses to Reviewer 2:**

*Claim to be the first space-time factorization with graph processing and comparisons to [A, B]:* To our best
knowledge, our work is unique by: a) proposing a message-passing, spatio-temporal graph model that incorporates
differently space and time information and works with unstructured video features, s.t. nodes are not associated with
distinct, semantic entities; b) our graph is recurrent in space and time, suited for online processing. It alternates
messages in time with those in space. [A, B] also use space-time separation on a graph, but models are not recurrent.
The nodes are associated with skeleton data or actors-object-scene info extracted with additional methods. Our model
uses unstructured features provided by a convnet backbone and nodes are not associated with specific entities (e.g.
objects or joints). This has an advantage: it permits independent end-to-end learning and inference, with no need
external detectors. We thank R2 for the references, which we will discuss in the final version.

*Comparisons to other works on Charades:* Please also see our response to R1 and R2 w.r.t tests on Charades.
Regarding comparisons to [B], we point out that our main task is activity recognition, whereas [B] tackles action
localization. Thus, direct comparison is not as trivial. Also, method in [C] focuses on other tasks, while results on
activity recognition, shown in supplementary material, are inferior to state of the art. We thank R2 for these recently
published references, which we will include and discuss in the paper.

*Ablation studies regarding message passing, region-split scheme and number of scales:*  In Section 3.1.1 we
present extensive ablation studies with different types of message passing based on MLPs, which validate the relevance
of recurrence with different processing over space and time. We agree with R2 that additional studies comparing to
simpler space processing in the form of linear graph convolutions will bring additional insights and we will include such
experiments in the final version. Also, we think that using objects as nodes is orthogonal to our approach of creating
nodes from fixed regions. We argue that such a fixed organization, independent of the output of external detectors is
more flexible and has certain advantages. It allows us to function independent of the exact number of objects in the
scene, which could change from one moment to the next. It also relieves us from needing to detect entities and then
match between nodes and entities. Thus, adapting our graph model to work with an external detector is indeed not
trivial. Instead, we directly compared our method to a top performing one that uses objects as nodes [33]. We also
performed ablation studies on the graph structure by varying the number of nodes and scales. We observed, for example,
that the model with 30 nodes (4 scales) obtains slightly lower results (by 0.28%) while being $1.6 \times$ slower than the
model with 14 nodes (3 scales). We will include such ablation studies on number of nodes and scales if accepted.

**Additional answers to Reviewer 3:**

We thank the reviewer for the helpful and positive comments. We will add the time comparisons to NL-I3D, as
shown in the Table above. Our RSTG-to-vec model is faster and more accurate than NL, while our top performer is
slightly slower. We will add and discuss these results in the final version.

[Meta-Review · NeurIPS 2019]

All reviewers agree that the submission should be accepted and appreciate the novelty, clarity and potential for impact. The authors are asked to please update the camera-ready with the suggestions from the reviewers, particularly wrt: - GPU memory requirements, and time comparisons, especially wrt to NL-I3D - additional clarification of claims and other points - extra references